# MODIFY TRAINING DIRECTION IN FUNCTION SPACE TO REDUCE GENERALIZATION ERROR

## ABSTRACT

To improve generalization performance by modifying the training dynamics, we present theoretical analyses of a modified natural gradient descent method in the neural network function space, leveraging the neural tangent kernel theory. Firstly, we provide an analytical expression for the function acquired through this modified natural gradient descent under the assumptions of an infinite network width limit and a Gaussian conditional output distribution. Subsequently, we explicitly derive the generalization error associated with the learned neural network function. By interpreting the generalization error as stemming from the distribution discrepancy between the training data and the true data, we propose a criterion for modification in the eigenspaces of the Fisher information matrix to reduce the generalization error bound. Through this approach, we establish that modifying the training direction of the neural network in function space leads to a reduction in generalization error. These theoretical results are also illustrated through numerical experiments. Additionally, we demonstrate the connections between this theoretical framework and existing results of generalization-enhancing methods.

## 1 INTRODUCTION

Neural networks have achieved impressive success in tackling various challenging tasks appeared in real world. However, understanding the generalization performance of neural networks remains a complex and intricate problem for researchers.

Many factors affect generalization error of a model, such as the structure of neural network, the datasets utilized, the optimization algorithm chosen for training. A modern neural network always possessed more than millions of parameters, resulting in highly complex parameter space that make it extremely challenging to analyze their generalization error. However, a clearer perspective emerges when considering in the function space, since neural network is devoted to approximate the true model in a function space rather than parameter space. Recently, the seminal work of [16] proved that in infinite width limit, the parameter-update based training dynamics can be converted to a differential dynamical system associated with Neural Tangent Kernel (NTK) in function space. But conventional gradient descent optimization algorithms such as Stochastic Gradient Descent (SGD) [9], RMSProp [37], Adam [21] are only operate directly in parameter space. Natural Gradient [3], which utilizes curvature information, is a reparameterization invariant gradient based optimization method which exhibits a strong connection with function space. In function space, the training dynamics of neural network can be interpreted as training in each eigenspace [36; 8]. Since different eigenspace associated with different spectrum contributes differently to the training dynamics[8] and consequently to the generalization error, there might exist an operation to modify the training dynamics in eigenspaces to enhance the generalization performance of the function learned. Building upon the aforementioned insights, we firstly propose an approximate explicit solution of an over-parameterized neural network trained by Modified natural gradient descent (Modified NGD) under specific assumptions. Based on the explicit solution, we interpret the generalization error as stemming from the distribution discrepancy between the training data and the true data, then propose a criterion for modification in the eigenspaces of the Fisher information matrix to reduce the generalization error bound.

Several methods have been proposed to enhance generalization performance, including gradient suppression in cross-domain-generalization [15] and self-distillation [39; 30]. Connections exist

between these methods and our theoretical framework, for these methods implicitly modify the eigenvalues of the Fisher information matrix in function space and consequently the training direction in the function space.

## 2 RELATED WORK

Since neural network is a complex system, whose generalization error is difficult to track, it is reasonable to simplify the case to more trackable and representative models such as kernel method. With kernel method, there are many impressive results on generalization error bounds [10; 6; 17; 26; 34; 8]. The classical results of Bartlett [6] proved that the generalization error bound of kernel method is positively correlated with the trace of the kernel. Jacot et al. [17] derived a risk estimator for kernel ridge regression. [26] derived a closed-form generalization error of kernel regression for teacher-student distillation framework. [34] revealed that the convergence rate of kernel method. And [8] decomposed the average generalization error into eigencomponents under the Mercer's condition.

The forward process of an infinite-width neural network can be described using a concept known as Neural Network Gaussian process (NNGP) [24]. Consequently, wide neural networks can be approximated by linear models [22]. Additionally, a well-established theoretical result of Jacot [16] has demonstrated that in infinite width limit, neural networks are dominated by a constant kernel referred to as the neural tangent kernel (NTK). This property allows for the application of results from kernel methods to wide neural networks. Notably, recent theoretical developments concerning the NTK have provided a rich foundation for analyzing generalization in the NTK regime [16; 5; 24; 12; 31; 22]. Numerous studies have investigated the impact of overparameterization on generalization error, revealing that overparameterization tends to lead to convergence toward flat minima [39; 20], and aids in escaping local minima [33]. Other perspectives view generalization as a form of compression [4] and neuron unit-wise capacity [23] for analyses. Based on these theoretical results, many researches have incorporated SGD within the NTK regime. For instance, [38] analyzed the training loss trajectory of SGD based on the spectrum of a loss operator, while various theoretical results of generalization error bounds of SGD have been derived in the context of the NTK regime [35; 11; 1; 25; 23].

Due to the high-dimensional complexity of the neural network parameter space, the effect of SGD in parameter space is not readily discernible. Natural Gradient Descent (NGD), firstly proposed by Amari et al. [3], takes into account curvature information in function space. Martens [28] established a connection between Fisher information matrix and Kullback-Leibler divergence in function space, demonstrating the reparameterization invariance of NGD. In NTK regime, Bernacchia et al. [7] derived an explicit expression for the convergence rate of NGD in deep linear neural networks. Rudner et al. [32] provided an analytical solution of NGD with linearization under infinite width limit. Additionally, Karakida et al. [19] established that, under specific conditions, existing approximate Fisher methods for NGD, such as K-FAC [29; 13], exhibit the same convergence properties as exact NGD.

In this paper, we leveraging the theoretical properties of NGD and NTK to provide an analytical solution for Modified NGD and derive an explicit expression for generalization error. Utilizing this expression, we modify the training direction of NGD in function space by modifying the Fisher information matrix to reduce the generalization error bound. These theoretical results are also illustrated through numerical experiments. Moreover, we demonstrate the connections between our theoretical framework and existing results of generalization-enhancing methods, such as [15; 39; 30].

## 3 PRELIMILARIES

### 3.1 PROBLEM SETUP

Suppose the distribution of data points and labels is $p_{\text{data}}(x, y)$, where $x \in \mathbb{R}^{d_{in}}, y \in \mathbb{R}$, the training set $\{(x_i, y_i)\}_{i=1}^N \sim \hat{p}_{\text{data}}(x, y)$, and the training data and the training label after vectorization is respectively $\mathcal{X} = (x_i)_{i=1}^N$ and $\mathcal{Y} = (y_i)_{i=1}^N$, then $\mathcal{X} \in \mathbb{R}^{N \cdot d_{in}}, \mathcal{Y} \in \mathbb{R}^N$. A fully connected neural network with $L$ layers whose width are respectively $d_l, l = 1, 2, \ldots, L$, can be expressed as:

$$f_\theta(x) = W^{(L)} \sigma \left( W^{(L-1)} \sigma \left( \ldots \sigma \left( W^{(1)} x + b^{(1)} \right) \ldots \right) + b^{(L-1)} \right) + b^{(L)} \tag{1}$$

where $\sigma$ is the element-wise activation function, $\theta = \left(W^{(1)}, W^{(2)}, \ldots, W^{(L)}, b^{(1)}, b^{(2)}, \ldots, b^{(L)}\right)$ is the weights of the network, $W^{(l)} \in \mathbb{R}^{d_{l-1} \times d_l}, b \in \mathbb{R}^{d_l}$ for $l = 1, 2, \ldots, L$, with $d_0 = d_{in}$ and $d_L = 1$.

The objective of training a neural network using a training set $(\mathcal{X}, \mathcal{Y})$ is to minimize the loss function with respect to the parameter $\theta$ within a parameter space $\Theta \subset \mathbb{R}^P$, where $P = \sum_{l=1}^{L}(d_{l-1} + 1)d_l$:

$$\min_{\theta \in \Theta} \mathcal{L}\left(f_\theta(\mathcal{X}), \mathcal{Y}\right). \tag{2}$$

In the following sections, we take $\mathcal{L}$ to be $L^2$ loss, i.e.

$$\mathcal{L}\left(f_\theta(\mathcal{X}), \mathcal{Y}\right) = \frac{1}{2N}\|f_\theta(\mathcal{X}) - \mathcal{Y}\|_2^2. \tag{3}$$

The generalization error, also known as expected risk, is defined as:

**Definition 1.** *Suppose the data distribution is $p_{\text{data}}(x, y)$, and corresponding marginal distributions are $p_{\text{data}}(x)$ and $p_{\text{data}}(y)$, then for a predictor $f$, which maps the input $x$ to the output $y$, the expected risk of $f$ w.r.t. $L^2$ loss is*

$$\mathcal{R}(f) = \mathbb{E}_{(x,y) \sim p_{\text{data}}(x,y)}\left[\|f_{\theta_0}(x) - y\|_2^2\right]. \tag{4}$$

## 3.2 Natural Gradient Descent

Let $\mathcal{H}$ be some function space, and $\mathcal{L}$ be a divergence, $f_\theta \in \mathcal{H}$ is a parameterized function, then the natural gradient under Kullback-Leibler divergence (KL divergence) of $\mathcal{L}(f_\theta, y)$ at point $\theta$ is defined as

$$\tilde{\nabla}_\theta \mathcal{L} = \boldsymbol{F}^{-1} \nabla_\theta \mathcal{L} \tag{5}$$

where $\boldsymbol{F} = \mathbb{E}_{x,y}\left[\nabla_\theta f_\theta(x, y) \nabla_\theta f_\theta(x, y)^\top\right]$ is the *Fisher information Matrix* of $f_\theta$.

Natural Gradient Descent (NGD), defined based on natural gradient is an algorithm with parameter update rule that

$$\Delta\theta_t = -\eta \tilde{\nabla}_\theta \mathcal{L}, \tag{6}$$

where $\eta$ is the learning rate.

## 3.3 Modified Natural Gradient Descent

We propose a noval natural gradient descent algorithm framework called *Modified Natural Gradient Descent* (Modified NGD).

Firstly, for a function $\varphi : \mathbb{R} \to \mathbb{R}$, we define the $\varphi$−transform of a diagonal matrix $\boldsymbol{\Lambda} = (\lambda_1, \ldots, \lambda_n)$ as $\boldsymbol{\Lambda}_\varphi = (\varphi(\lambda_1), \ldots, \varphi(\lambda_n))$.

Then we define the $\varphi$−transform of a matrix $\boldsymbol{A}$, which apply the transformation $\varphi$ to its non-zero singular values as

$$\boldsymbol{A}_\varphi = \boldsymbol{U}\begin{pmatrix}\boldsymbol{\Lambda}_\varphi & \boldsymbol{0}\end{pmatrix}\boldsymbol{V}^\top, \quad \text{where} \quad \boldsymbol{A} = \boldsymbol{U}\begin{pmatrix}\boldsymbol{\Lambda} & \boldsymbol{0}\end{pmatrix}\boldsymbol{V}^\top \quad \text{is the SVD decomposition of } \boldsymbol{A}. \tag{7}$$

Apply $\varphi$−transform to the Fisher information matrix $\boldsymbol{F}$. With $\boldsymbol{F}_\varphi$, we define the Modified NGD as follows:

$$\begin{aligned}
\tilde{\nabla}_\theta \mathcal{L} &= \boldsymbol{F}_\varphi^{-1} \nabla_\theta \mathcal{L}, \\
\Delta\theta_t &= -\eta \tilde{\nabla}_\theta \mathcal{L}
\end{aligned} \tag{8}$$

where $\eta$ is the learning rate.

In the following sections, we will first derive the analytical solution for Modified NGD. Subsequently, we will establish, within the NTK regime, that training with Modified NGD using an appropriate transformation $\varphi$ yields a lower generalization error compared to conventional NGD.

# 4 MAIN RESULTS

## 4.1 ANALYTICAL SOLUTION OF MODIFIED NGD

Let us begin by outlining the principal assumptions made in this paper:

**Assumption 1.** *For a data point $x$ and a network function $f$, we assume the output conditional probability $\tilde{p}(y|f(x))$ is Gaussian:*

$$\tilde{p}(y|f(x)) = \frac{1}{\sqrt{2\pi}\sigma_0}e^{\frac{(y-f(x))^2}{2\sigma_0^2}}. \tag{9}$$

**Assumption 2.** *(**NTK Regime Assumption**) The neural network is with linear output layer and Lipschitz activation function. The wights of neural network are initialized using He initialization [14]. And the width of the layers of the neural network tends to infinity, as indicated in the network expression 1 that :*

$$n_l \to \infty, \; l = 1, 2, \ldots, L - 1. \tag{10}$$

**Assumption 3.** *The neural network is with linear output layer and Lipschitz activation function. The wights of neural network are initialized with He initialization [14]. And the neural network is overparameterized, which is typically described as having layer widths exceeding a polynomial order of the training sample size[2; 5; 18; 24], that is*

$$\min_{l \in \{1, \ldots, L\}} d_l > poly(N). \tag{11}$$

*which generally means that $P \gg N$.*

**Assumption 4.** *The neural tangent kernel at initialization is positive definite, or equivalently, the following term is positive definite:*

$$\nabla_\theta f_{\theta_0}(\mathcal{X})\nabla_\theta f_{\theta_0}(\mathcal{X})^\top$$

**Assumption 5.** *For simplicity and without loss of generality, we take the transform $\varphi$ as the following form:*

$$\varphi(\xi) = \begin{cases} 0, & \text{if } c(\xi) = \text{True} \\ \xi, & \text{if } c(\xi) = \text{False}. \end{cases} \tag{12}$$

*For some criterion $c$ that takes binary values.*

Since the empirical Fisher $\tilde{\boldsymbol{F}}(\theta_t) \in \mathbb{R}^{P \times P}$ is given by

$$\begin{aligned} \tilde{\boldsymbol{F}}(\theta_t) &= \frac{1}{N}\mathbb{E}_{\tilde{p}(y|f(\mathcal{X}))}\left[\nabla_\theta \log \tilde{p}(y|f(\mathcal{X}))\nabla_\theta \log \tilde{p}(y|f(\mathcal{X}))^\top\right] \\ &= \frac{1}{N}\mathbb{E}_{\tilde{p}(y|f(\mathcal{X}))}\left[\nabla_\theta f(\mathcal{X})^\top \nabla_f \log \tilde{p}(y|f(\mathcal{X}))\nabla_f \log \tilde{p}(y|f(\mathcal{X}))^\top \nabla_\theta f(\mathcal{X})\right]. \end{aligned} \tag{13}$$

Under Assumption 1, the empirical Fisher 32 can be written as

$$\tilde{\boldsymbol{F}}(\theta_t) = \frac{1}{N\sigma_0^2}\nabla_\theta f(\mathcal{X})^\top \nabla_\theta f(\mathcal{X}). \tag{14}$$

Under Assumption 2, the neural network has a linearization expression as:

$$f_{\theta_t}(x) = f_{\theta_0}(x) + \nabla_\theta f_{\theta_0}(x)(\theta_t - \theta_0). \tag{15}$$

Under the linearization, the Jacobian matrix of $f_\theta$ remains constant. Therefore, the NTK and the Fisher are both constant during training. Denoting the Jacobian matrix of $f_{\theta_t}$ evaluated on training data points $\mathcal{X}$ at $\theta_t$ as $\boldsymbol{J}_t(\mathcal{X})$, and abbrevating $\boldsymbol{J}_0(\mathcal{X})$ for $\boldsymbol{J}$ unless otherwise specified.

Under Assumption 3, equation 15 is an approximation for the network function and the following analyses are based on this linear approximation. We can apply SVD decomposition to Jacobian matrix $\boldsymbol{J}$:

$$\boldsymbol{J} = \boldsymbol{U}\begin{pmatrix} \boldsymbol{\Lambda} & \boldsymbol{0}_{N,P-N} \end{pmatrix}\boldsymbol{V}^T \tag{16}$$

where $\boldsymbol{U} \in \mathbb{R}^{N \times N}$, $\boldsymbol{V} \in \mathbb{R}^{P \times P}$ are both orthogonal matrices, that is $\boldsymbol{U}\boldsymbol{U}^\top = \boldsymbol{U}^\top\boldsymbol{U} = \boldsymbol{I}_N$, $\boldsymbol{V}\boldsymbol{V}^\top = \boldsymbol{V}^\top\boldsymbol{V} = \boldsymbol{I}_P$, and $\boldsymbol{\Lambda} = diag(\lambda_1, \ldots, \lambda_N)$ with $\lambda_1 \geq \cdots \geq \lambda_N > 0$. Thus, we have the NTK $\mathbf{K}_t$ and empirical Fisher information matrix $\tilde{\boldsymbol{F}}(\theta_t)$ that

$$\mathbf{K}_t = \mathbf{K}_0 = \boldsymbol{J}\boldsymbol{J}^\top = \boldsymbol{U}\boldsymbol{\Lambda}^2\boldsymbol{U}^\top,$$

$$\tilde{\boldsymbol{F}}(\theta_t) = \tilde{\boldsymbol{F}}(\theta_0) = \frac{1}{N\sigma_0^2}\boldsymbol{J}^\top\boldsymbol{J} = \frac{1}{N\sigma_0^2}\boldsymbol{V}\begin{pmatrix} \boldsymbol{\Lambda}^2 & \boldsymbol{0} \\ \boldsymbol{0} & \boldsymbol{0} \end{pmatrix}\boldsymbol{V}^\top. \tag{17}$$

The modification operation under assumption 5 on $\boldsymbol{\Lambda}^2$ can be written as

$$(\boldsymbol{\Lambda}^2)_\varphi = diag\left(\varphi\left(\lambda_1^2\right), \ldots, \varphi\left(\lambda_N^2\right)\right) = \boldsymbol{\Lambda}^2\boldsymbol{I}_{c(\boldsymbol{\Lambda}^2)}. \tag{18}$$

where $\boldsymbol{I}_{c(\boldsymbol{\Lambda}^2)}$ is a modified identity whose positions being set zeros if the criterion $c$ for the corresponding positions of $\boldsymbol{\Lambda}^2$ hold.

Thus the pseudo inverse of empirical Fisher information matrix evaluated at $\theta_0$ can be wriiten as

$$\left(\tilde{\boldsymbol{F}}_\varphi\right)^\dagger \triangleq \left(\tilde{\boldsymbol{F}}_\varphi(\theta_0)\right)^\dagger = N\sigma_0^2\boldsymbol{V}\begin{pmatrix} \left[(\boldsymbol{\Lambda}^2)_\varphi\right]^\dagger & \boldsymbol{0} \\ \boldsymbol{0} & \boldsymbol{0} \end{pmatrix}\boldsymbol{V}^\top = N\sigma_0^2\boldsymbol{V}\begin{pmatrix} \boldsymbol{\Lambda}^{-2}\boldsymbol{I}_{c(\boldsymbol{\Lambda}^2)} & \boldsymbol{0} \\ \boldsymbol{0} & \boldsymbol{0} \end{pmatrix}\boldsymbol{V}^\top. \tag{19}$$

Then we can derive the analytical solution of Modified NGD with training set $\mathcal{X}$ and $\mathcal{Y}$.

**Theorem 1.** *Under Assumptions 1, 3, 4, and linearization approximation equation 15, with $L^2$ loss and the decomposition equation 17, the solution of Modified NGD 8 trained on $\mathcal{X}$ and $\mathcal{Y}$ for time $T$ has prediction $f_{\theta_T}(x)$ on the test point $x \sim p_{\text{data}}(x)$, which can be expressed analytically as:*

$$f_{\theta_t}(x) = f_{\theta_0}(x) - \frac{1}{\sigma_0^2}\left(1 - e^{-\eta N\sigma_0^2 t}\right)\boldsymbol{J}(x)\left(\tilde{\boldsymbol{F}}_\varphi\right)^\dagger\nabla_{\theta_0}\mathcal{L}(f(\mathcal{X}), \mathcal{Y}). \tag{20}$$

The proof of Theorem 1 can be found in the **Appendix**.

**Remark 1.** *From the proof of Theorem 1, we can see that*

$$\frac{\partial f_{\theta_t}(\mathcal{X})}{\partial t} = -\eta\boldsymbol{J}\left(\tilde{\boldsymbol{F}}_\varphi\right)^\dagger\boldsymbol{J}^\top\left(f_{\theta_t}(\mathcal{X}) - \mathcal{Y}\right),$$

*and*

$$\frac{\partial f_{\theta_t}(x)}{\partial t} = -\eta\boldsymbol{J}(x)\left(\tilde{\boldsymbol{F}}_\varphi\right)^\dagger\boldsymbol{J}^\top\left(f_{\theta_t}(\mathcal{X}) - \mathcal{Y}\right).$$

*The modification on Fisher information matrix will change the training direction of $f_t$ in function space, and consequently influence the convergence point of $f_t$. Hence, selecting an appropriate modification can potentially reduce the generalization error of the convergence function trained using Modified NGD.*

Theorem 1 gives the neural network function trained by Modified NGD algorithm for time $T$. As the convergence theory of NG algorithm [7], we claim that the network function trained by Modified NGD converges as $T \to \infty$.

**Corollary 1.** *Under the same assumptions as Theorem 1, the network function trained by Modified NGD converges to $f_{\theta_\infty}(x)$ as $T \to \infty$,*

$$f_{\theta_\infty}(x) = \lim_{T \to \infty} f_{\theta_T}(x) = f_{\theta_0}(x) - \frac{1}{\sigma_0^2}\boldsymbol{J}(x)\left(\tilde{\boldsymbol{F}}_\varphi\right)^\dagger\nabla_{\theta_0}\mathcal{L}(f(\mathcal{X}), \mathcal{Y}). \tag{21}$$

Based on the solutions given by Theorem 1 and Corollary 1, we can analyze the generalization error of the network function trained by Modified NGD on training set. In the next subsection, we will derive the generalization error of the convergence network function and corresponding criterion for modification to reduce this generalization error bound.

## 4.2 GENERALIZATION ERROR BOUND REDUCTION

For the convergence network function $f_{\theta_\infty}$ in equation 21 trained by Modified NGD, we can derive the generalization error of the convergence network function and corresponding criterion for modification to reduce this generalization error bound.

**Theorem 2.** *Under the same assumptions as Theorem 1, the expected risk of $f_{\theta_\infty}$ trained by Modified NGD in Corollary 1 can be expressed as:*

$$
\begin{aligned}
\mathcal{R}(f_{\theta_\infty}) =& \mathbb{E}_x\left[(f_{\theta_0}(x)-y)^2\right] - \frac{2}{\sigma_0^2}\mathbb{E}_{x,y}\left[\nabla_{\theta_0}\mathcal{L}(f(x),y)^\top\right]\left(\tilde{\boldsymbol{F}}_\varphi\right)^\dagger \nabla_{\theta_0}\mathcal{L}(f(\mathcal{X}),\mathcal{Y}) \\
&+ \frac{1}{\sigma_0^2}\nabla_{\theta_0}\mathcal{L}(f(\mathcal{X}),\mathcal{Y})^\top\left(\tilde{\boldsymbol{F}}_\varphi\right)^\dagger \boldsymbol{F}^\star\left(\tilde{\boldsymbol{F}}_\varphi\right)^\dagger \nabla_{\theta_0}\mathcal{L}(f(\mathcal{X}),\mathcal{Y}).
\end{aligned}
\tag{22}
$$

*where $F^\star = \mathbb{E}_{x,y}\left[\frac{1}{\sigma_0^2}\boldsymbol{J}(x)^\top\boldsymbol{J}(x)\right]$ is the Fisher information matrix on the true data distribution.*

*Denote $\alpha(x,y) = f_{\theta_0}(x) - y$, $\alpha(\mathcal{X},\mathcal{Y}) = f_{\theta_0}(\mathcal{X}) - \mathcal{Y}$, and $\boldsymbol{U} = (u_1 \ \ldots \ u_N)$. Then with defining the criterion $c(\lambda_i^2)$, where $\lambda_i > 0$, to be True when the following inequality holds*

$$
\frac{\lambda_i\left(\boldsymbol{V}^\top(\boldsymbol{F}^\star)^\dagger \mathbb{E}_{x,y}\left[\nabla_{\theta_0}\mathcal{L}(f(x),y)\right]\right)_i}{\sigma_0^2 u_i^\top \alpha(\mathcal{X},\mathcal{Y})} < \frac{1}{2}
\tag{23}
$$

*the Modified NGD will reduce the generalization error bound.*

The proof of Theorem 2 can be found in the **Appendix**.

**Remark 2.** *From the proof of Theorem 2, it shows that the ideal modified Fisher should satisfy the condition*

$$
\left(\tilde{\boldsymbol{F}}_{\varphi^\star}\right)^\dagger \nabla_{\theta_0}\mathcal{L}(f(\mathcal{X}),\mathcal{Y}) = (\boldsymbol{F}^\star)^\dagger \mathbb{E}_{x,y}[\nabla_{\theta_0}\mathcal{L}(f(x),y)].
\tag{24}
$$

*On the left-hand side is the modified training direction in parameter space, while the right-hand side represents the training direction on the true data distribution. This condition implies that the modified training direction should closely align with the training direction based on the true data distribution. Consequently, the training direction in function space is adjusted to closely match the true model function.*

**Remark 3.** *If the distribution of training data is consistent with the distribution of true data, then by the law of large number, when the sample size $N$ is large enough, we have*

$$
\begin{aligned}
\tilde{\boldsymbol{F}} &\approx \boldsymbol{F}^\star, \\
\nabla_{\theta_0}\mathcal{L}(f(\mathcal{X}),\mathcal{Y}) &\approx \mathbb{E}_{x,y}[\nabla_{\theta_0}\mathcal{L}(f(x),y)].
\end{aligned}
\tag{25}
$$

*In this case, when $\tilde{\boldsymbol{F}}_{\varphi^\star} = \tilde{\boldsymbol{F}}$, it serves as the optimal candidate for equation 24, signifying that no modification is necessary in the absence of distribution discrepancy. And the criterion*

$$
\begin{aligned}
\frac{\lambda_i\left(\boldsymbol{V}^\top(\boldsymbol{F}^\star)^\dagger \mathbb{E}_{x,y}\left[\nabla_{\theta_0}\mathcal{L}(f(x),y)\right]\right)_i}{\sigma_0^2 u_i^\top \alpha(\mathcal{X},\mathcal{Y})} &\approx \frac{\lambda_i(V^\top\left(\frac{1}{N\sigma_0^2}\boldsymbol{J}^\top\boldsymbol{J}\right)^\dagger \frac{1}{N}\boldsymbol{J}^\top\alpha(\mathcal{X},\mathcal{Y}))_i}{\sigma_0^2 u_i^\top \alpha(\mathcal{X},\mathcal{Y})} \\
&= \frac{\lambda_i\left(\lambda_i^{-1}u_i^\top\alpha(\mathcal{X},\mathcal{Y})\right)}{u_i^\top\alpha(\mathcal{X},\mathcal{Y})} = 1 < \frac{1}{2}.
\end{aligned}
\tag{26}
$$

*This observation also recommends no modification of the Fisher matrix when the distributions of the training data and the true data are consistent, aligning with empirical claims.*

## 5 NUMERICAL EXPERIMENTS

This section aims to illustrate our theoretical results of Modified NGD. Specifically, based on the theoretical criterion for modification, the Modified NGD can reduce the generalization error compared with ordinary NGD and NGD with modifications with other criteria. It's important to note that the primary goal in conducting these experiments is to provide empirical support for our theoretical results, rather than proposing a practical algorithm[1].

---

[1]All codes, data and results are available in the **Supplementary Materials**. More details can be found in the **Appendix**.

Due to the high dimension of Fisher, we have refrained from employing complex network architectures. With the dicussions of the discrepancy bounds of NTK regime and general neural network [16; 5; 32], our theoretical and numerical results is possible to be generalized to general deep neural networks (DNN).

We illustrate our theoretical results through two numerical experiments: the first one employs a synthetic dataset to validate the effectiveness of our Modified NGD with derived theoretical criterion 23 with distribution discrepancy of dataset; the second one is implemented with HTRU2 dataset, a realistic dataset which describes a sample of pulsar candidates collected during the High Time Resolution Universe Survey [27], to present the generality of our therotical results. We use a MLP model with three hidden layers of $2^8$, $2$ and $2^{12}$ neurons perspectively, with He initialization [14] and MSE loss as the loss function. $\sigma_0^2$ in Gaussian conditional output distribution assumption 1 is set as 0.01 for all experiments. For all experiments in this paper, when we perform different optimization algorithms, except the optimization algorithm itself, all the other settings such as learning rate are same. Modified NGD uses validation set for the true distribution computation in the criterion 23 to decide which direction to be modified.

SETUP

**Synthetic Dataset Experiments** To generate the synthetic data for a function fitting task, we firstly draw samples uniformly from interval $[0, 1)$, then split the samples to training set with 256 samples, validation set with 64 samples and test set with 64 samples, and apply perturbation to the training set:

$$x \to xe^{-\frac{(1-x)^2}{\sigma^2}} \tag{27}$$

with different perturbation factor of $\sigma^2$. This perturbation results in different distributions of training set and test set, while the distributions of the validation set and test set remaining same.

The object of the model is to fit the following function:

$$f^\star(x) = \cos x \sin x. \tag{28}$$

The first tuple of experiments involves two algorithms: NGD and Modified NGD, on training sets with different perturbations. For robustness and reliability, we conduct each experiment using 20 random seeds, and the results are reported on the average. We implements the numerical experiments with the perturbation factors $\sigma^2$ being set as: 10, 5, and 1.

The second tuple of experiments aim to demonstrate that the advantages of Modified NGD are not the result of arbitrary eigenvalue truncation in the Fisher information matrix, but the effectiveness of the theoretical criterion 23. In this tuple of experiments, we employ with four algorithms: NGD, Modified NGD, NGD with small eigenvalues being cut and NGD with large eigenvalues being cut. For the latter two algorithms, the eigenvalues of the Fisher information matrix are modified with specific criteria: small eigenvalues are set to zero while preserving the larger ones in one case, and large eigenvalues are set to zero while preserving the smaller ones in the other. Importantly, we ensure the number of Fisher's non-zero eigenvalues being same after modification for the latter three algorithms. The training set is sujected to a perturbation factor of $\sigma^2 = 1$.

For all experiments on synthetic dataset, the initial learning rate is set as 1 with learning rate half decay and train for 500 epochs.

**HTRU2 Dataset Experiments** The HTRU2 dataset contains total 17,898 examples, 16,259 spurious examples caused by RFI/noise, and 1,639 real pulsar examples. These examples have all been checked by human annotators. Each candidate is described by 8 continuous variables. We implemented two algorithms: NGD and Modified NGD on HTRU2 dataset.

The first experiment is implemented with HTRU2 dataset splited to training set, validation set and test set of ratio $8 : 1 : 1$, and the three sets are of similar distribution that negative examples : positive examples $\approx 0.91 : 0.09$.

In the second experiments, the distribution of training set is made different from validation set and test set, where negative examples : positive examples $= 0.875 : 0.125$ in training set while $0.75 : 0.25$ in validation set and test set.

For all experiments on HTRU2 dataset, the initial learning rate is set as 1 with learning rate half decay and train for 200 epochs.

RESULTS

**Synthetic Dataset Experiments** The numerical results of the first tuple of experiments on synthetic dataset are depicted in Figure 1 and figure 2. In figure 1, figure 1a, figure 1b and figure 1c are the test loss results trained on training data perturbed with the perturbation factor of $\sigma^2 = 10, 5$ and 1, perspectively. These results illustrate that Modified NGD consistently exhibits a lower generalization error compared to NGD across various levels of perturbation in the training data. In the plots, a line represents the mean on random seeds and the envelope around it reflects 0.3 times standard deviation, the start point of the plots is the 50th epoch. Notably, as $\sigma^2$ decreases, indicating a decrease in the homogeneity between the training and test sets, a degradation in NGD's performance arises compared to Modified NGD. Figure 2 provides insight into the trend of the difference between NGD and Modified NGD at the convergence point. It becomes evident that this difference decreases as the distribution of the training and test sets approaches similarity. Figure 3 presents the results of the

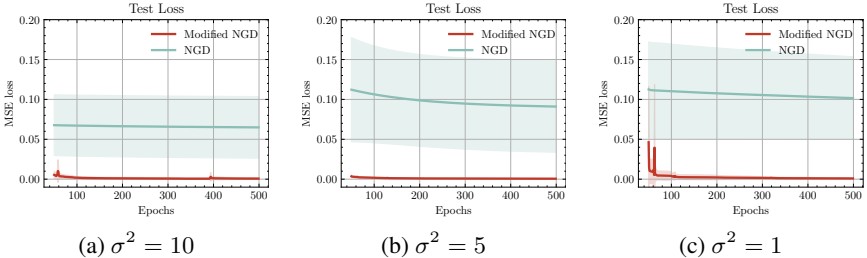

(a) $\sigma^2 = 10$  (b) $\sigma^2 = 5$  (c) $\sigma^2 = 1$

Figure 1: The test loss of NGD and Modified NGD with different degrees of perturbation.

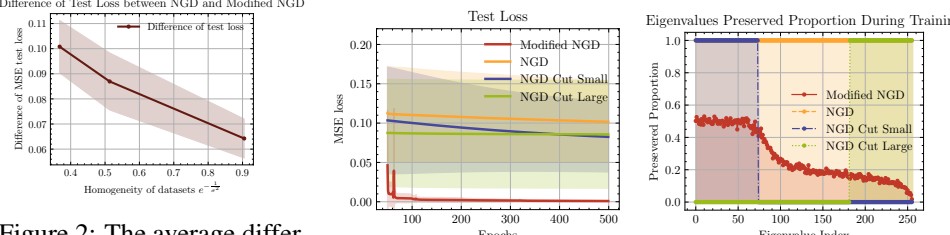

Figure 2: The average difference of test loss of NGD and Modified NGD in the last 10 epochs.

Figure 3: The results of algorithms with different modification criteria.

second tuple of experiments conducted on synthetic dataset with the perturbation factor of $\sigma^2 = 1$. The test loss of these four algorithms are shown in left subfigure of figure 3, the start point of the plot is the 50th epoch. The experimental results demonstrate that modifying eigenvalues through the criteria of "cutting small" or "cutting large" is indeed effective in reducing the generalization error. However, our proposed criterion 23 consistently outperforms these approaches, thus validating our theoretical results. The right subfigure of figure 3 illustrates the proportion of eigenvalues of Fisher information matrix being preserved during 500 training epochs. It can be obeserved that almost all eigenvalues would be modified in some epochs, and smaller one are more likely to be modified.

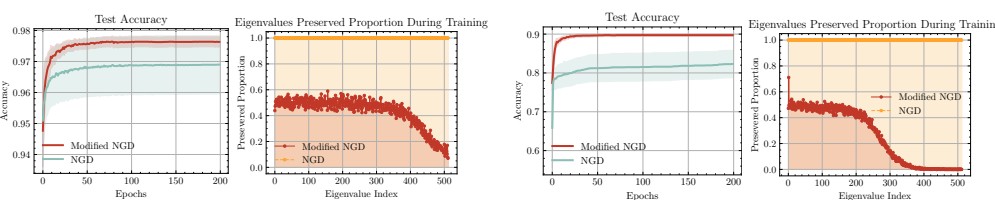

Figure 4: Results on HTRU2 dataset.  Figure 5: Results on perturbed HTRU2 dataset.

**HTRU2 Dataset Experiments** The numerical results of the first and second experiment on HTRU2 dataset are depicted in Figure 4 and figure 5, respectively. The left subfigure of figure 4 illustrates the test accuracy of NGD and Modified NGD with more homogeneous training set and test set, while the left subfigure of figure 5 with more heterogeneous training set and test set, where a line represents the mean on random seeds and the envelope around it reflects 0.3 times standard deviation, the start point of the plots is the 0th epoch. The right subfigures of figure 4 and figure 5 illustrate the proportion of eigenvalues of Fisher information matrix being preserved during 200 training epochs of these two algorithms. From the results, we can observe that when the training set and test set share a similar distribution, the test accuracy of both NGD and Modified NGD remains close, with a difference of less than $1\%$. However, as the distribution discrepancy between them increases, this difference escalates to approximately $8\%$. Additionally, the proportion of preserved eigenvalues indicates that more eigenvalues would be modified by our criterion 23 when confronted with lerger distribution discrepancy.

## 6 CONNECTION WITH EXISTING GENERALIZATION-ENHANCING METHODS

Since the machine learning achieved good performance on a lot of tasks, several algorithms aiming for enhancing the generalization performance were proposed. There are some connections between our theoretical framework and thier methods. In the following, we give two examples.

**Cross domain generalization** Zeyi Huang et al. [15] proposed an intuitive algorithm to enhance the performance of cross-domain generalization by cutting the largest components of the gradient $\nabla_\theta \mathcal{L}(f(\mathcal{X}), \mathcal{Y})$, which can be derived as modifying the singular values of the Jacobian matrix as:

$$\nabla_{\theta,\text{Cut lagrest}}\mathcal{L}(f(\mathcal{X}), \mathcal{Y}) = \boldsymbol{J}'\left(f(\mathcal{X}) - \mathcal{Y}\right), \tag{29}$$

where $\boldsymbol{J}'$ is the transformed Jacobian matrix. Therefore, we can define a Modified NGD with modified Fisher as

$$\tilde{\boldsymbol{F}}_\varphi \triangleq \boldsymbol{J}'\left(\boldsymbol{J}^\top\right)^\dagger. \tag{30}$$

for some transformation $\varphi$ that may be in a more general transformation class than the class in Assumption 5. With this transformation, Modified NGD will possesses the same training directions as their method in the training dynamics.

**Self distillation** Self distillation is a post-training method. Mobahi et al. [30] shows that self distillation amplifies regularization effect at each distillation round, which make the eigenvalues of the Gram matrix of the kernel of the regularizer evolve. And after several distillation rounds, the new corresponding kernel's Gram matrix possesses smaller eigenvalues, thus enhances the generalization performance. They shown in [30] that the solution of the regularized optimization problem after $t$ rounds distillation is equivalent to the solution of a modified kernel without distillation

$$f_t^\star(x) = \mathbf{g}_x^{\top\dagger}\left(c_0 \boldsymbol{I} + \mathbf{G}^\dagger\right)^{-1}\mathcal{Y}. \tag{31}$$

where $\mathbf{g}$ is the Green function of the regularizer ,$c_i$ are the regulaization parameters, and $\mathbf{G}^\dagger$ is the Gram matrix of the modified Green function. Compared [30] with the solution of Modified NGD, we can observe that the modified Gram matrix in [30] has the similar role of the modified Fisher matrix in Modified NGD. And Mobahi et al. [30] proved that the eigenvalues $\lambda_k^\dagger$ of modified Gram matrix is descending as $t$ increasing. Therefore, in our framework, self distillation employs a mild modification on training directions in function space introduced by the kernel.

## 7 CONCLUSION

We firstly presented a Modified NGD framework and proceed to derive an analytical expression for the function trained by this Modified NGD. Based on this solution, we explicitly computed the generalization error of the learned neural network function and proposed a criterion to decide the directions to be modified. We established theoretical results and implemented numerical experiments to verify that modifying the training direction of the neural network in function space leads to a reduction in the generalization error bound. Furthermore, We demonstrate the connections between this theoretical framework and existing results of generalization-enhancing methods.

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

# A   APPENDIX

## THE DERIVATION OF EMPIRICAL FISHER INFORMATION MATRIX

Recall Assumption 1. Since the empirical Fisher $\tilde{F}(\theta_t) \in \mathbb{R}^{P \times P}$ is given by

$$
\begin{aligned}
\tilde{\boldsymbol{F}}(\theta_t) &= \frac{1}{N} \mathbb{E}_{\tilde{p}(y|f(\mathcal{X}))} \left[ \nabla_\theta \log \tilde{p}(y|f(\mathcal{X})) \nabla_\theta \log \tilde{p}(y|f(\mathcal{X}))^\top \right] \\
&= \frac{1}{N} \mathbb{E}_{\tilde{p}(y|f(\mathcal{X}))} \left[ \nabla_\theta f(\mathcal{X})^\top \nabla_f \log \tilde{p}(y|f(\mathcal{X})) \nabla_f \log \tilde{p}(y|f(\mathcal{X}))^\top \nabla_\theta f(\mathcal{X}) \right].
\end{aligned}
\tag{32}
$$

If we assume the output probability is Gaussian,

$$
\nabla_f \log \tilde{p}(y|f(\mathcal{X})) = \nabla_f \left( \frac{(y - f(\mathcal{X})^\top (y - f(\mathcal{X})))}{2\sigma_0^2} \right) = \frac{y - f(\mathcal{X})}{\sigma_0^2},
\tag{33}
$$

Then, the empirical Fisher 32 can be writer as

$$
\begin{aligned}
\tilde{\boldsymbol{F}}(\theta_t) &= \frac{1}{N\sigma_0^4} \mathbb{E}_{\tilde{p}(y|f(\mathcal{X}))} \left[ \nabla_\theta f(\mathcal{X})^\top (y - f(\mathcal{X}))(y - f(\mathcal{X}))^\top \nabla_\theta f(\mathcal{X}) \right] \\
&= \frac{1}{N\sigma_0^4} \nabla_\theta f(\mathcal{X})^\top \mathbb{E}_{\tilde{p}(y|f(\mathcal{X}))} \left[ (y - f(\mathcal{X}))(y - f(\mathcal{X}))^\top \right] \nabla_\theta f(\mathcal{X}) \\
&= \frac{1}{N\sigma_0^4} \nabla_\theta f(\mathcal{X})^\top \sigma_0^2 I \nabla_\theta f(\mathcal{X}) \\
&= \frac{1}{N\sigma_0^2} \nabla_\theta f(\mathcal{X})^\top \nabla_\theta f(\mathcal{X}).
\end{aligned}
\tag{34}
$$

## PROOF OF THEOREM 1

*Proof.* Firstly, we derivate the solution of Modified NGD on training set $(\mathcal{X}, \mathcal{Y})$.

The training dynamics of Modified NGD in function space on training set can be write as:

$$
\frac{\partial f_{\theta_t}(\mathcal{X})}{\partial t} = \nabla_\theta f_{\theta_t}(\mathcal{X}) \frac{\partial \theta_t(\mathcal{X})}{\partial t} = -\eta \boldsymbol{J} \left( \tilde{\boldsymbol{F}}_\varphi \right)^\dagger \boldsymbol{J}^\top (f_{\theta_t}(\mathcal{X}) - \mathcal{Y}).
\tag{35}
$$

Since

$$
\boldsymbol{J} \left( \tilde{\boldsymbol{F}}_\varphi \right)^\dagger \boldsymbol{J}^\top = N\sigma_0^2 \boldsymbol{U} \boldsymbol{I}_{c(\boldsymbol{\Lambda}^2)} \boldsymbol{U}^\top
\tag{36}
$$

$$\frac{\partial f_{\theta_t}(\mathcal{X})}{\partial t} = -\eta N \sigma_0^2 \boldsymbol{U} \boldsymbol{I}_{c(\boldsymbol{\Lambda}^2)} \boldsymbol{U}^\top \left( f_{\theta_t}(\mathcal{X}) - \mathcal{Y} \right), \tag{37}$$

we can analytically solve this ODE by

$$\begin{aligned}
f_{\theta_t}(\mathcal{X}) &= \mathcal{Y} + e^{-\eta N \sigma_0^2 \boldsymbol{U} \boldsymbol{I}_{c(\boldsymbol{\Lambda}^2)} \boldsymbol{U}^\top t} \left( f_{\theta_0}(\mathcal{X}) - \mathcal{Y} \right) \\
&= \mathcal{Y} + \left[ \boldsymbol{I} + \left( e^{-\eta N \sigma_0^2 t} - 1 \right) \boldsymbol{U} \boldsymbol{I}_{c(\boldsymbol{\Lambda}^2)} \boldsymbol{U}^\top \right] \left( f_{\theta_0}(\mathcal{X}) - \mathcal{Y} \right), \quad \forall t \in [0, T].
\end{aligned} \tag{38}$$

After that, let us foucs on the function dynamics on test point $x \sim p_{\text{data}}(x)$. Recall the expression of $f_{\theta_t}(\mathcal{X})$ in equation 38, we have

$$\begin{aligned}
\frac{\partial f_{\theta_t}(x)}{\partial t} &= -\eta \boldsymbol{J}(x) \left( \tilde{\boldsymbol{F}}_\varphi \right)^\dagger \boldsymbol{J}^\top \left( f_{\theta_t}(\mathcal{X}) - \mathcal{Y} \right) \\
&= -\eta \boldsymbol{J}(x) \left( \tilde{\boldsymbol{F}}_\varphi \right)^\dagger \boldsymbol{J}^\top \left[ \boldsymbol{I} + \left( e^{-\eta N \sigma_0^2 t} - 1 \right) \boldsymbol{U} \boldsymbol{I}_{c(\boldsymbol{\Lambda}^2)} \boldsymbol{U}^\top \right] \left( f_{\theta_0}(\mathcal{X}) - \mathcal{Y} \right) \\
&= -\eta N \sigma_0^2 \boldsymbol{J}(x) \boldsymbol{V} \begin{pmatrix} \boldsymbol{\Lambda}^{-1} \boldsymbol{I}_{c(\boldsymbol{\Lambda}^2)} \\ \boldsymbol{0} \end{pmatrix} \boldsymbol{U}^\top \left[ \boldsymbol{I} + \left( e^{-\eta N \sigma_0^2 t} - 1 \right) \boldsymbol{U} \boldsymbol{I}_{c(\boldsymbol{\Lambda}^2)} \boldsymbol{U}^\top \right] \left( f_{\theta_0}(\mathcal{X}) - \mathcal{Y} \right) \\
&= -\eta N \sigma_0^2 e^{-\eta N \sigma_0^2 t} \boldsymbol{J}(x) \boldsymbol{V} \begin{pmatrix} \boldsymbol{\Lambda}^{-1} \boldsymbol{I}_{c(\boldsymbol{\Lambda}^2)} \\ \boldsymbol{0} \end{pmatrix} \boldsymbol{U}^\top \left( f_{\theta_0}(\mathcal{X}) - \mathcal{Y} \right).
\end{aligned} \tag{39}$$

Integrad by $t$ in the two sides of this equation, we get

$$\begin{aligned}
f_{\theta_t}(x) &= f_{\theta_0}(x) - \left( 1 - e^{-\eta N \sigma_0^2 t} \right) \boldsymbol{J}(x) \boldsymbol{V} \begin{pmatrix} \boldsymbol{\Lambda}^{-1} \boldsymbol{I}_{c(\boldsymbol{\Lambda}^2)} \\ \boldsymbol{0} \end{pmatrix} \boldsymbol{U}^\top \left( f_{\theta_0}(\mathcal{X}) - \mathcal{Y} \right) \\
&= f_{\theta_0}(x) - \frac{1}{N \sigma_0^2} \left( 1 - e^{-\eta N \sigma_0^2 t} \right) \boldsymbol{J}(x) \left( \tilde{\boldsymbol{F}}_\varphi \right)^\dagger \boldsymbol{J}^\top \left( f_{\theta_0}(\mathcal{X}) - \mathcal{Y} \right) \\
&= f_{\theta_0}(x) - \frac{1}{\sigma_0^2} \left( 1 - e^{-\eta N \sigma_0^2 t} \right) \boldsymbol{J}(x) \left( \tilde{\boldsymbol{F}}_\varphi \right)^\dagger \nabla_{\theta_0} \mathcal{L}(f(\mathcal{X}), \mathcal{Y}).
\end{aligned} \tag{40}$$

This solution holds for $\forall t \in [0, T]$. In particular, it holds for $t = T$, which concludes the proof. $\square$

PROOF OF THEOREM 2

*Proof.* Recall the definition of expected risk and the expression of $f_\infty$ that

$$f_{\theta_t}(x) = f_{\theta_0}(x) - \boldsymbol{J}(x) \boldsymbol{V} \begin{pmatrix} \boldsymbol{\Lambda}^{-1} \boldsymbol{I}_{c(\boldsymbol{\Lambda}^2)} \\ \boldsymbol{0} \end{pmatrix} \boldsymbol{U}^\top \left( f_{\theta_0}(\mathcal{X}) - \mathcal{Y} \right). \tag{41}$$

We have

$$\begin{aligned}
\mathcal{R}(f_{\theta_\infty}) &= \mathbb{E}_{(x,y) \sim p_{\text{data}}(x,y)} \left[ \left( f_{\theta_\infty}(x) - y \right)^2 \right] \\
&= \mathbb{E}_x \left[ \left( f_{\theta_0}(x) - y - \boldsymbol{J}(x) \boldsymbol{V} \begin{pmatrix} \boldsymbol{\Lambda}^{-1} \boldsymbol{I}_{c(\boldsymbol{\Lambda}^2)} \\ \boldsymbol{0} \end{pmatrix} U^\top \left( f_{\theta_0}(\mathcal{X}) - \mathcal{Y} \right) \right)^2 \right] \\
&= \mathbb{E}_x \left[ \left( f_{\theta_0}(x) - y \right)^2 \right] \\
&\quad - 2 \mathbb{E}_x \left[ \left( f_{\theta_0}(x) - y \right) \boldsymbol{J}(x) \boldsymbol{V} \begin{pmatrix} \boldsymbol{\Lambda}^{-1} \boldsymbol{I}_{c(\boldsymbol{\Lambda}^2)} \\ \boldsymbol{0} \end{pmatrix} U^\top \left( f_{\theta_0}(\mathcal{X}) - \mathcal{Y} \right) \right] \\
&\quad + \mathbb{E}_x \left[ \left( \boldsymbol{J}(x) \boldsymbol{V} \begin{pmatrix} \boldsymbol{\Lambda}^{-1} \boldsymbol{I}_{c(\boldsymbol{\Lambda}^2)} \\ \boldsymbol{0} \end{pmatrix} U^\top \left( f_{\theta_0}(\mathcal{X}) - \mathcal{Y} \right) \right)^2 \right] \\
&\triangleq T_1 - T_2 + T_3.
\end{aligned} \tag{42}$$

The first term on the right side $T_1$ is

$$T_1 = \mathbb{E}_x \left[ \left( f_{\theta_0}(x) - y \right)^2 \right] \tag{43}$$

For the second term on the right side $T_2$, we have

$$
\begin{aligned}
T_2 &= 2\mathbb{E}_{x,y}\left[\left(f_{\theta_0}(x) - y\right)\boldsymbol{J}(x)\boldsymbol{V}\begin{pmatrix}\boldsymbol{\Lambda}^{-1}\boldsymbol{I}_{c(\boldsymbol{\Lambda}^2)} \\ \boldsymbol{0}\end{pmatrix}\boldsymbol{U}^\top\left(f_{\theta_0}(\mathcal{X}) - \mathcal{Y}\right)\right] \\
&= 2\mathbb{E}_{x,y}\left[\left(f_{\theta_0}(x) - y\right)\boldsymbol{J}(x)\right]\boldsymbol{V}\begin{pmatrix}\boldsymbol{\Lambda}^{-1}\boldsymbol{I}_{c(\boldsymbol{\Lambda}^2)} \\ \boldsymbol{0}\end{pmatrix}\boldsymbol{U}^\top\left(f_{\theta_0}(\mathcal{X}) - \mathcal{Y}\right) \\
&= \frac{2}{\sigma_0^2}\mathbb{E}_{x,y}\left[\nabla_{\theta_0}\mathcal{L}(f(x), y)^\top\right]\left(\tilde{\boldsymbol{F}}_\varphi\right)^\dagger\nabla_{\theta_0}\mathcal{L}(f(\mathcal{X}), \mathcal{Y}).
\end{aligned}
\tag{44}
$$

The third term $T_3$ can be similarly rewritten as

$$
\begin{aligned}
T_3 &= \mathbb{E}_x\left[\left(J(x)\boldsymbol{V}\begin{pmatrix}\boldsymbol{\Lambda}^{-1}\boldsymbol{I}_{c(\boldsymbol{\Lambda}^2)} \\ \boldsymbol{0}\end{pmatrix}\boldsymbol{U}^\top\left(f_{\theta_0}(\mathcal{X}) - \mathcal{Y}\right)\right)^2\right] \\
&= \alpha(\mathcal{X}, \mathcal{Y})^\top\boldsymbol{U}\begin{pmatrix}\boldsymbol{\Lambda}^{-1}\boldsymbol{I}_{c(\boldsymbol{\Lambda}^2)} & \boldsymbol{0}\end{pmatrix}\boldsymbol{V}^\top\mathbb{E}_x\left[\boldsymbol{J}(x)^\top\boldsymbol{J}(x)\right]\boldsymbol{V}\begin{pmatrix}\boldsymbol{\Lambda}^{-1}\boldsymbol{I}_{c(\boldsymbol{\Lambda}^2)} \\ \boldsymbol{0}\end{pmatrix}\boldsymbol{U}^\top\alpha(\mathcal{X}, \mathcal{Y}) \\
&= \frac{1}{\sigma_0^2}\nabla_{\theta_0}\mathcal{L}(f(\mathcal{X}), \mathcal{Y})^\top\left(\tilde{\boldsymbol{F}}_\varphi\right)^\dagger\boldsymbol{F}^\star\left(\tilde{\boldsymbol{F}}_\varphi\right)^\dagger\nabla_{\theta_0}\mathcal{L}(f(\mathcal{X}), \mathcal{Y}).
\end{aligned}
\tag{45}
$$

Therefore, the generaliztion error can be written as:

$$
\begin{aligned}
\mathcal{R}(f_{\theta_\infty}) =& \mathbb{E}_x\left[\left(f_{\theta_0}(x) - y\right)^2\right] - \frac{2}{\sigma_0^2}\mathbb{E}_{x,y}\left[\nabla_{\theta_0}\mathcal{L}(f(x), y)^\top\right]\left(\tilde{\boldsymbol{F}}_\varphi\right)^\dagger\nabla_{\theta_0}\mathcal{L}(f(\mathcal{X}), \mathcal{Y}) \\
&+ \frac{1}{\sigma_0^2}\nabla_{\theta_0}\mathcal{L}(f(\mathcal{X}), \mathcal{Y})^\top\left(\tilde{\boldsymbol{F}}_\varphi\right)^\dagger\boldsymbol{F}^\star\left(\tilde{\boldsymbol{F}}_\varphi\right)^\dagger\nabla_{\theta_0}\mathcal{L}(f(\mathcal{X}), \mathcal{Y}).
\end{aligned}
\tag{46}
$$

Now let us regard the term $\left(\tilde{\boldsymbol{F}}_\varphi\right)^\dagger\nabla_{\theta_0}\mathcal{L}(f(\mathcal{X}), \mathcal{Y})$ as a free variable $\xi$, then consider the quadratic function:

$$
\begin{aligned}
\mathcal{R}(f_{\theta_\infty})(\xi) =& \frac{1}{\sigma_0^2}\left(\xi^\top\boldsymbol{F}^\star\xi - 2\mathbb{E}_{x,y}\left[\nabla_{\theta_0}\mathcal{L}(f(x), y)^\top\right]\xi\right. \\
&\left.+ \mathbb{E}_x\left[\nabla_{\theta_0}\mathcal{L}(f(x), y)^\top(F^\star)^\dagger\nabla_{\theta_0}\mathcal{L}(f(x), y)^\top\right]\right)
\end{aligned}
\tag{47}
$$

Since $\boldsymbol{F}^\star$ is positive semi-definite, the above function obtains its minimum at

$$
\xi^\star = (\boldsymbol{F}^\star)^\dagger\mathbb{E}_{x,y}\left[\nabla_{\theta_0}\mathcal{L}(f(x), y)\right].
\tag{48}
$$

which is equivalently

$$
\left(\tilde{\boldsymbol{F}}_\varphi\right)^\dagger\nabla_{\theta_0}\mathcal{L}(f(\mathcal{X}), \mathcal{Y}) = (\boldsymbol{F}^\star)^\dagger\mathbb{E}_{x,y}\left[\nabla_{\theta_0}\mathcal{L}(f(x), y)\right]
\tag{49}
$$

Write the lefthands more explicitly, we have

$$
\begin{aligned}
\sigma_0^2\boldsymbol{V}\begin{pmatrix}\left((\boldsymbol{\Lambda}^2)_\varphi\right)^\dagger & \boldsymbol{0} \\ \boldsymbol{0} & \boldsymbol{0}\end{pmatrix}\boldsymbol{V}^\top\boldsymbol{V}\begin{pmatrix}\boldsymbol{\Lambda} \\ \boldsymbol{0}\end{pmatrix}\boldsymbol{U}^\top\alpha(\mathcal{X}, \mathcal{Y}) &= (\boldsymbol{F}^\star)^\dagger\mathbb{E}_{x,y}\left[\nabla_{\theta_0}\mathcal{L}(f(x), y)\right] \\
\begin{pmatrix}\boldsymbol{\Lambda}\left((\boldsymbol{\Lambda}^2)_\varphi\right)^\dagger \\ \boldsymbol{0}\end{pmatrix}\boldsymbol{U}^\top\alpha(\mathcal{X}, \mathcal{Y}) &= \frac{1}{\sigma_0^2}\boldsymbol{V}^\top(\boldsymbol{F}^\star)^\dagger\mathbb{E}_{x,y}\left[\nabla_{\theta_0}\mathcal{L}(f(x), y)\right]
\end{aligned}
\tag{50}
$$

For the $i$-th component ($1 \le i \le N$) of the above equation, we have

$$
\lambda_i\left(\varphi(\lambda_i^2)\right)^\dagger u_i^\top\alpha(\mathcal{X}, \mathcal{Y}) = \frac{1}{\sigma_0^2}\left(\boldsymbol{V}^\top(\boldsymbol{F}^\star)^\dagger\mathbb{E}_{x,y}\left[\nabla_{\theta_0}\mathcal{L}(f(x), y)\right]\right)_i.
\tag{51}
$$

Notice that there may be no proper transformation $\varphi$ in our transformation class in Assumption 5. So we denote the transformation satisfying equation 51 as the ideal transformation $\varphi^\star$.

Define the $\boldsymbol{F}^\star$-semi-norm for vector $\xi$ as

$$
\|\xi\|_{\boldsymbol{F}^\star} \triangleq \xi^\top\boldsymbol{F}^\star\xi.
\tag{52}
$$

We can rewrite $\mathcal{R}(f_{\theta_\infty})(\xi)$ with respect to $\mathcal{R}(f_{\theta_\infty})(\xi^\star)$:

$$
\begin{aligned}
\mathcal{R}(f_{\theta_\infty})(\xi) =& \mathcal{R}(f_{\theta_\infty})\left((\xi - \xi^\star) + \xi^\star\right) \\
=& \frac{1}{\sigma_0^2}\Big(\left((\xi - \xi^\star) + \xi^\star\right)^\top \boldsymbol{F}^\star \left((\xi - \xi^\star) + \xi^\star\right) - 2\mathbb{E}_{x,y}\left[\nabla_{\theta_0}\mathcal{L}(f(x),y)^\top\right]\left((\xi - \xi^\star) + \xi^\star\right) \\
& + \mathbb{E}_x\left[\nabla_{\theta_0}\mathcal{L}(f(x),y)^\top (F^\star)^\dagger \nabla_{\theta_0}\mathcal{L}(f(x),y)^\top\right]\Big) \\
=& \frac{1}{\sigma_0^2}(\xi - \xi^\star)^\top \boldsymbol{F}^\star (\xi - \xi^\star) + \mathcal{R}(f_{\theta_\infty})(\xi^\star) \\
& + \frac{2}{\sigma_0^2}(\xi^\star)^\top \boldsymbol{F}^\star(\xi - \xi^\star) - \frac{2}{\sigma_0^2}\mathbb{E}_{x,y}\left[\nabla_{\theta_0}\mathcal{L}(f(x),y)^\top\right](\xi - \xi^\star) \\
=& \frac{1}{\sigma_0^2}\|\xi - \xi^\star\|_{\boldsymbol{F}^\star} + \mathcal{R}(f_{\theta_\infty})(\xi^\star).
\end{aligned}
$$

$$(53)$$

To minimize $\mathcal{R}(f_{\theta_\infty})(\xi)$ with $\xi$ with corresponding transformation $\varphi$ in the transformation class in Assumption 5, it is equivalently to minimize $\|\xi - \xi^\star\|_{\boldsymbol{F}^\star}$ in corresponding transformation class.

By the norm equivalence theorem in finite dimensional vector space, we have

$$
\begin{aligned}
\|\xi - \xi^\star\|_{\boldsymbol{F}^\star} \leq& C_1\|\xi - \xi^\star\|_2 \\
\leq& C_1 N\sigma_0^2\|\boldsymbol{V}\|_2^2 \|\mathbb{E}_{x,y}\left[\nabla_{\theta_0}\mathcal{L}(f(x),y)\right]\|_2 \\
& \cdot \left\|\begin{pmatrix} \left(\varphi(\lambda_1^2)\right)^\dagger - \left(\varphi^\star(\lambda_1^2)\right)^\dagger & & \\ & \ddots & \\ & & \left(\varphi(\lambda_N^2)\right)^\dagger - \left(\varphi^\star(\lambda_N^2)\right)^\dagger \end{pmatrix}\right\|_2 \\
=& C_2 \sum_{i=1}^N \left[\left(\varphi(\lambda_i^2)\right)^\dagger - \left(\varphi^\star(\lambda_i^2)\right)^\dagger\right]^2.
\end{aligned}
$$

$$(54)$$

The above bound determined the generalization error bound of Modified NGD. To minimize the last term is equivalently to minimize each eigencomponent $\left[\left(\varphi(\lambda_i^2)\right)^\dagger - \left(\varphi^\star(\lambda_i^2)\right)^\dagger\right]^2$ for $1 \leq i \leq N$.

Recall our form of transformation $\varphi$ with criterion $c$ in Assumption 5. If we want the generalization error bound with Modified NGD got minimized, it is directly to set $c(\lambda_i^2)$ as

$$
c(\lambda_i^2) = \left(\left[\left(\varphi^\star(\lambda_i^2)\right)^\dagger\right]^2 < \left[\lambda_i^{-2} - \left(\varphi^\star(\lambda_i^2)\right)^\dagger\right]^2\right)
$$

$$(55)$$

Take the expression of $\varphi^\star(\lambda_i^2)$ in equation 51, we have

$$
\begin{aligned}
c(\lambda_i^2) =& \left(\left[\left(\varphi^\star(\lambda_i^2)\right)^\dagger\right]^2 < \left[\lambda_i^{-2} - \left(\varphi^\star(\lambda_i^2)\right)^\dagger\right]^2\right) \\
=& \left(\left[\frac{\left(\boldsymbol{V}^\top (\boldsymbol{F}^\star)^\dagger \mathbb{E}_{x,y}\left[\nabla_{\theta_0}\mathcal{L}(f(x),y)\right]\right)_i}{\sigma_0^2 \lambda_i u_i^\top \alpha(\mathcal{X},\mathcal{Y})}\right]^2 < \left[\lambda_i^{-2} - \frac{\left(\boldsymbol{V}^\top (\boldsymbol{F}^\star)^\dagger \mathbb{E}_{x,y}\left[\nabla_{\theta_0}\mathcal{L}(f(x),y)\right]\right)_i}{\sigma_0^2 \lambda_i u_i^\top \alpha(\mathcal{X},\mathcal{Y})}\right]^2\right) \\
=& \left(\left[\frac{\lambda_i\left(\boldsymbol{V}^\top (\boldsymbol{F}^\star)^\dagger \mathbb{E}_{x,y}\left[\nabla_{\theta_0}\mathcal{L}(f(x),y)\right]\right)_i}{\sigma_0^2 u_i^\top \alpha(\mathcal{X},\mathcal{Y})}\right]^2 < \left[1 - \frac{\lambda_i\left(\boldsymbol{V}^\top (\boldsymbol{F}^\star)^\dagger \mathbb{E}_{x,y}\left[\nabla_{\theta_0}\mathcal{L}(f(x),y)\right]\right)_i}{\sigma_0^2 \lambda_i u_i^\top \alpha(\mathcal{X},\mathcal{Y})}\right]^2\right) \\
=& \left(\frac{\lambda_i\left(\boldsymbol{V}^\top (\boldsymbol{F}^\star)^\dagger \mathbb{E}_{x,y}\left[\nabla_{\theta_0}\mathcal{L}(f(x),y)\right]\right)_i}{\sigma_0^2 u_i^\top \alpha(\mathcal{X},\mathcal{Y})} < \frac{1}{2}\right).
\end{aligned}
$$

$$(56)$$

Notice that the $\lambda_i$, $1 \leq i \leq N$, are singular values thus non-negative, no confusion would arise in the definition of $c(\lambda_i^2)$. $\qquad\square$

NUMERICAL EXPERIMENTS

This section aims to illustrate our theoretical results of Modified NGD, that is, based on the theoretical criterion for modification, the Modified NGD can reduce the generalization error compared with ordinary NGD and NGD with modification on other directions.

Due to the high dimension of Fisher, all of our experiments are implemented on a two layers MLP (Multi-Layer Perceptron) with synthetic data. However, with the dicussions of the discrepancy bounds of NTK regime and general neural network [16; 5; 32], our theoretical and numerical results can be generalized to general DNN.

We illustrate our theoretical results by two numerical experiments[2]: the first one is implemented with synthetic dataset to varify that our Modified NGD with derived theoretical criterion 23 is effiective to correct the training direction for better generalization with respect to distribution discrepancy of dataset; the second one is implemented with HTRU2 dataset, a realistic dataset which describes a sample of pulsar candidates collected during the High Time Resolution Universe Survey [27], to present the generality of our therotical results. We use a MLP model with three hidden layers of $2^8$, 2 and $2^{12}$ neurons perspectively, with He initialization [14] and MSE loss as the loss function. $\sigma_0^2$ in Gaussian conditional output distribution assumption 1 is set as $0.01$ for all experiments. For all experiments in this paper, when we perform different optimization algorithms: NGD, Modified NGD, NGD with small eigenvalues being cut and NGD with large eigenvalues being cut, except the optimization algorithm itself, all the other settings such as learning rate are same. Modified NGD uses validation set for the true distribution computation in the criterion 23 to decide which direction to be modified. NGD with small eigenvalues being cut and NGD with large eigenvalues being cut modify the eigenvalues of Fisher with criteria that small eigenvalues being set to zeros whlie remaining larger ones and large eigenvalues being set to zeros whlie remaining smaller ones, respectively. To compute the Fisher information matrix and criterion 23 on training set and validation set, we perform SVD decomposition to the Jacobian matrices in the training procedure. For the stability of training, for all optimization algorithms used in this paper, we set a threshold to suppress too large eigenvalues of the inverse Fisher. The threshold is taken as $1e3$, and the eigenvalues greater than it will be set as the value of the threshold. The Fisher information matrix is updated at each epoch for the beginning 25 epochs, and updated once per 10 epochs after that. The Fisher information matrix is updated with all training samples, while the criterion is computed with all validation samples in the synthetic dataset experiments. For the first experiment of HTRU2 dataset, the Fisher information matrix is updated with randomly sampled 512 training samples, while the criterion is computed with randomly sampled 512 validation samples. For the second experiment of HTRU2 dataset, the Fisher information matrix is updated with all training samples, while the criterion is computed with all validation samples.

**Synthetic Dataset Experiments** To generate the synthetic data for a function fitting task, we firstly draw samples uniformly from interval $[0, 1)$, then split the samples to training set with 256 samples, validation set with 64 samples and test set with 64 samples, and apply perturbation to the training set:

$$x \rightarrow x e^{-\frac{(1-x)^2}{\sigma^2}} \tag{57}$$

with different perturbation factor of $\sigma^2$. This perturbation results in different distributions of training set and test set, while the distributions of the validation set and test set remaining same.

The object of the model is to fit the following function:

$$f^\star(x) = \cos x \sin x. \tag{58}$$

The first tuple of experiments is implemented with two algorithms: NGD and Modified NGD, on training sets with different perturbations. We run each experiments for 20 random seed, and the results are reported on the average of different random seeds. We implements the numerical experiments for different degrees of perturbation with the mean of perturbed data changing roughly equally, where the perturbation factors $\sigma^2$ are set to be: 10, 5, and 1.

The second tuple of experiments aim to demonstrate that the benifit of Modified NGD is not due to casually cutting eigenvalues of Fisher information matrix, and the effectiveness of the theoretical criterion 23. The second tuple of experiments is implemented with four algorithms: NGD, Modified

---

[2]All codes, data and results are available in the **Supplementary Materials**. More details can be found in the **Appendix**.

NGD, NGD with small eigenvalues being cut and NGD with large eigenvalues being cut. For the later three algorithms, we ensure the number of Fisher's non-zero eigenvalues of them being same after modification. The training set is with perturbation factor of $\sigma^2 = 1$.

For all experiments on synthetic dataset, the initial learning rate is set as 1 with learning rate half decay, batch size is set as 256, and train for 500 epochs.

**HTRU2 Dataset Experiments** The HTRU2 dataset contains total 17,898 examples, 16,259 spurious examples caused by RFI/noise, and 1,639 real pulsar examples. These examples have all been checked by human annotators. Each candidate is described by 8 continuous variables. We implemented two algorithms: NGD and Modified NGD on HTRU2 dataset.

The first experiment is implemented with HTRU2 dataset splited to training set, validation set and test set of ratio $8 : 1 : 1$, and the three sets are of similar distribution that negative examples : positive examples $\approx 0.91 : 0.09$. Here we sample the three sets with Python functional Numpy.random.shuffle() to achieve similar distributions.

In the second experiments, the distribution of training set is made different from validation set and test set, where negative examples : positive examples $= 0.875 : 0.125$ in training set while $0.75 : 0.25$ in validation set and test set. Here we first use Python functional Numpy.random.shuffle() for positive examples and negative examples, perspectively. Then we split the positive set and negative set to three subsets, perspectively. For training set, we randomly sample 64 positive examples from the first subset of positive set and 448 negative examples from the first subset of the negative set, then the negative examples : positive examples is $0.875 : 0.125$ in training set; For validation set, we randomly sample 128 positive examples from the second subset of positive set and 128 negative examples from the second subset of the negative set, then the negative examples : positive examples is $0.75 : 0.25$ in validation set; For test set, we randomly sample 128 positive examples from the third subset of positive set and 128 negative examples from the third subset of the negative set, then the negative examples : positive examples is $0.75 : 0.25$ in test set.

For all experiments on HTRU2 dataset, the initial learning rate is set as 1 with learning rate half decay, batch size is set as 256, and train for 200 epochs.

