# OpenReview forum: "Modify Training Direction in Function Space to Reduce Generalization Error"
_ICLR.cc/2024/Conference — Submitted to ICLR 2024_

### Official Review · Reviewer_KgvZ · 2023-10-23

**Soundness:** 2 fair
**Presentation:** 3 good
**Contribution:** 3 good
**Rating:** 5
**Confidence:** 2

**Summary:**

This paper proposes a modified natural gradient descent (NGD) method by modifying the eigenvalues of the Fisher information matrix to improve the generalization performance.
The main results are two-folded: a convergence analysis of the modified NGD and a generalization bound of the convergence model.
Specifically, in the NTK (lazy training) regime, this paper gives an explicit expression for the final convergence model under the modified NGD method.
Then this paper designs a modification method to reduce the generalization error of the final convergence model by adjusting the eigenvalues of the Fisher information matrix.

**Strengths:**

* The paper is written clearly and the ideas have been presented in the proper order.
* The proposed modification method for the eigenvalues of the Fisher information matrix is interesting, and  may offer several possible future directions for the optimization algorithm designs to improve the generalization performance.

**Weaknesses:**

* The Gaussian likelihood assumption 1 seems to be restrictive in the sense that it only holds for Gaussian data, is it possible to relax this assumption?
* The NTK regime assumption 2 also restricts the model to be a random feature model, which can simplify the analysis but overparameterization requirement is also restrictive.
* The optimization result (Theorem 1) is based on the convergence result in [32], which limits the novelty.
* It would be more convincing if more numerical experiments are conducted to show the effectiveness of the modified NGD method. For example, the image classification task on the CIFAR10/CIFAR100 dataset.


Some minor issues:
* the last step in equation (26) should be $1 > \frac{1}{2}$.
* the left hand side of equation (41) should be $f_{\theta_{\infty}}(x)$.



[32] Tim GJ Rudner, Florian Wenzel, Yee Whye Teh, and Yarin Gal. The natural neural tangent kernel: Neural network training dynamics under natural gradient descent. *4th workshop on Bayesian Deep Learning (NeurIPS 2019), 2019*.

**Questions:**

I have some questions on the details of the modified NGD method:
* The criterion (23) in Theorem 2 seems to be dependent on the distribution of the data $(x,y)$. Does it mean that we can only operate the modified NGD if we know the data distribution?
Because if we do not know the data distribution, and we use the empirical Fisher information matrix and the empirical gradient to approximate their expected values, then by equation (26) we do not need to modify the eigenvalues, which returns to NGD.
* Is the criterion (23) operated only at initialization for $\theta_0$? During training, we go back to NGD or still use the modified NGD?
* I do not understand why by setting $c(\lambda_i^2)$ as equation (55) can minimize the generalization bound of the modified NGD?
From my understanding, if we want to minimize the last term of equation (54), then we want to find $\varphi$ such that $\varphi(\lambda_i^2)^{\dagger} = \varphi^*(\lambda_i^2)^{\dagger}$, but why (55) can achieve this?

---

### Official Review · Reviewer_ZET7 · 2023-10-27

**Soundness:** 2 fair
**Presentation:** 2 fair
**Contribution:** 2 fair
**Rating:** 3
**Confidence:** 3

**Summary:**

The paper considers Natural Gradient Descent (NGD) training of neural networks, and then proposes a way to modify the Fisher information matrix so that the resulting Modified NGD achieves lower generalization error at the end of the training. In the theoretical part, the authors consider The network in the NTK regime in order to have constant Jacobian (and therefore NTK and Fisher) during training. Then, the authors additionally assume that the conditional distribution of the network output is isotropic Gaussian so that the Fisher matrix can be diagonalized by the same orthogonal transformation appearing in Jacobian SVD. Overall, this allows to analytically solve NGD dynamics. Then, the authors take the final network prediction at $t=\infty$ from the obtained solution and derive an upper bound (eq. (54)) on the generalization error. Finally, the optimal criterion within the considered class (12) by minimizing the derived upper bound.

In the experimental part, the authors compare their method with basic NGD, as well as two other NGD modifications with the same number of removed eigenvalues but chosen according to a simpler criterion based on their magnitude.

**Strengths:**

Overall, the paper asks interesting questions but answers them in a not very precise and systematic manner, so it is hard to single out strong points in the theoretical analysis.

However, the experiments shown on Figures 1-5 demonstrate very good performance of the proposed modified NGD algorithm compared to the basic NGD. In the case of a distribution shift between train and test data, it seems there is a potential in the proposed approach to improve generalization by comparing Fisher matrices calculated on train and validation data. In this regard, are there some intersections between your approach and paper [Canatar et al.,  2021](https://proceedings.neurips.cc/paper/2021/file/691dcb1d65f31967a874d18383b9da75-Paper.pdf)?

**Weaknesses:**

Overall, the paper feels rushed, with several typos in different places. This is not critical, but there are more conceptual concerns.

**The role of NGD.** When focusing on the main goal of the paper, as given by the title, the authors analyze the generalization performance of the final predictor $f_{\theta_\infty}$ given by (21) with modification $\varphi$ given within the class described by eq. (7) and (12). However, such a predictor simply corresponds to a complete fit of training residual $\alpha(\mathcal{X},\mathcal{Y})$ projected on a set of eigenspaces selected according to criterion $c$. From this perspective, Modified NGD seems to be only one of many possibilities (and a fairly complicated one) for obtaining such a predictor. For example, first projecting the residual on the selected eigenspaces and then training basic Gradient descent till convergence would produce the same predictor. This raises a question of the actual role of NGD in the analysis and results provided in the paper.

**Implication of final training dataset size.** The authors treat quite lightly the difference between quantities related to finite training size and those related to (presumably) infinite population distribution. For example, the equality (24) seems to be impossible to hold. As can be seen from its other form in eq. (50), the left-hand side can have only $N$ non-zero entries, while all entries of the right-hand side are presumably non-zero. This is just one implication of the broader issue of learning population quantities from finite number of observations, which is one of the main technical difficulties in references [8,10,17,26] provided in the manuscript and in some other works studying generalization error of kernel methods.

**The upper bound.** Theorem 2 presents the modification $\varphi$ that improves not the true error but its upper bound (54) derived in the appendix. In this regard, it would be reasonable to discuss this upper bound in the main text so there is more understanding of what is actually minimized. For example, I am not able to tell whether this bound based on the equivalence of norms in finite-dimensional spaces is typical in analysis of generalization error, and the authors do not give any references or comments about it. Also, I did not understand the transition from the first to the second line in eq. (54) - I would expect an additional term with the contribution from components of $\xi^\star$ orthogonal to directions with non-zero singular values of $\mathbf{J}$.

**Differences between theoretical and experimental settings.** The experimental and theoretical settings are quite different. Most importantly, in theoretical settings, the network is in NTK regime with constant Jacobian, while the experiments are performed with realistic finite networks with Jacobian evolving through training. Because of this, the authors regularly update Fisher during training. This aspect, often termed Feature Learning, is one of the main challenges to NTK theory, and it would be good to address it more clearly in the experiment design to make the relation to the theory more transparent.

Smaller remarks and typos which do not significantly affect the quality of the paper
- $1/N$ factor is missing in eq. (35). This is the reason for the not-very natural $N$ factor in the exponent $e^{-\eta N \sigma_0^2 t}$ in eq. (20).
- It seems strange that predictor $f$ (not the conditional probability $\widetilde{p}(y|x)$) depends on output $y$ in the definition of the Fisher matrix (just below eq. (5)).
- Theorem 1 gives the solution for (continuous) gradient flow algorithm, but formula (8) describes (discrete) gradient descent. It would be better to explicitly mention that the theorem is for gradient flow.

**Questions:**

It would be most interesting to address the concerns described in the above section. There are also a few smaller questions.

Theory related questions:
- Assumptions 2 and 3 are similar and differ in whether the width is infinite or just very large. Is this distinction important at some point of the paper?
- For the true data distribution $p_\mathrm{data}(x,y)$, are outputs $y$ deterministic when conditioned on an input $x$? Proof of theorem 2 assumes it - see eq. (42) - but it seems to be not mentioned in the text of the main paper or appendix.

Experiment related questions:
- In Figure 5, Modified and basic NGD start from significantly different accuracy values. But should not they have the same starting point, with the difference being only in the gradient updates?
- Is it possible to extend plots in figures 1. and 3a.? Otherwise, from the current plots it looks like almost no dynamics is happening.
- Is it possible to provide some baseline method for modification of Fisher matrix, so it will be possible to evaluate the effectiveness of the method proposed in the paper?

---

### Official Review · Reviewer_6uvN · 2023-10-29

**Soundness:** 3 good
**Presentation:** 3 good
**Contribution:** 2 fair
**Rating:** 3
**Confidence:** 4

**Summary:**

This paper proposed a way to improve the generalization bound of NTK by natural gradient method. The authors proposed a modified natural gradient method by clipping the singular values of the Fisher information matrix (which is used in natural gradient). The authors then derived an analytical expression for the modified natural gradient descent dynamics. The authors further derived a generalization bound with the learned network. The authors show that if the singular values is clipped in an appropriate way, the generalization error can decrease.

**Strengths:**

The authors is able to establish connection between natural gradient method and NTK and further show that it is able to improve generalization is novel and such result is previously less well-known to the deep learning community. I also appreciate the result that the modified natural gradient method is able to improve generalization.

**Weaknesses:**

Although I feel the connection is less well-known, overall, it feels like the entire idea is to apply something already well-established in kernel method to neural tangent kernel, which is a special kind of kernel. Another point I want to point out is that the neural network used in practice has been shown in various scenarios that it doesn't behave like a kernel. Thus, it is not clear whether the improved generalization from modified NGD can still be applied to neural networks not in NTK regime. With that, I feel the contribution of this work is somewhat limited. In the experiment section, the authors compare modified NGD to NGD. As a reader, I am also curious about the comparison of modified NGD and NGD to GD.

**Questions:**

1. The natural gradient method is developed under KL divergence, why it is applicable here (since you are using $L^2$ loss)?

**Details Of Ethics Concerns:**

None.

---

### Official Review · Reviewer_vsfs · 2023-11-05

**Soundness:** 2 fair
**Presentation:** 2 fair
**Contribution:** 2 fair
**Rating:** 3
**Confidence:** 3

**Summary:**

The authors propose a modification to the natural gradient descent algorithm. This modification is made by setting some singular values of the empirical Fisher information matrix to zero, based on satisfaction of a criterion. They derive the criterion in order to reduce the generalization error, under the NTK assumption. They show some proof-of-concept numerical results where their modification leads to lower out-of-distribution test errors.

**Strengths:**

Clearly defining the contributions

**Weaknesses:**

My main concern is that the results are limited in scope (to NTK, Gaussian data-likelihood) and hence not sufficiently novel and broadly applicable.

1. Under the NTK assumption, the dynamics are linear and hence it is analytically easy to derive the results in Theorem 2. None of the proof outlines are given.
1. Any modification to the gradient does change the functional gradient of the loss function. This is not a unique novelty of this paper.
Providing more intuition on the modifications for out-of-distribution generalization can enhance the results.

1. Figure captions could be more informative of what the inferences are.

**Questions:**

1. Is the criterion in theorem 2 just derived from stipulating that generalization error is lower than generalization error of NGD?
2. Does Remark 3 mean that modified NGD is the same as normal NGD when the test and training error are from the same distribution? Or is it only asymptotically true, as $N \to \infty$? Also, there is a $1/2 > 1$ in Remark 3.

Please spellcheck -- there are several spelling errors (e.g. wights) and also grammatical errors.

---

### Meta-Review · Area_Chair_2wPF · 2023-12-06

**Metareview:**

The paper considers natural-gradient descent on the parameters of a neural network, and aims to modify the descent direction to improve generalization. The modification entails setting some singular values of the Fisher information matrix to zero. In small-scale experiments, the method can lead to lower test error.

The paper tackles an interesting an important problem, but the are several weaknesses with the current paper. It relies on strong assumptions (NTK, Gaussianity) and shows only small-scale experiments.  The reviewers provide many constructive criticisms, which hopefully helps the authors to improve their work.

I recommend to reject the paper, but encourage the authors to improve the work by taking the reviewers' feedback into account.

**Justification For Why Not Higher Score:**

Many weaknesses were raised by the reviewers (see meta-review), to which the authors did not respond.

**Justification For Why Not Lower Score:**

N/A

---

### Decision · Program_Chairs · 2024-01-16

Reject